

**Lower functional redundancy in "narrow" than "broad" functions in global soil**
**metagenomics**
Huaihai Chen[a,*], Kayan Ma[a], Yu Huang[a], Jiajiang Lin[b], Christopher W. Schadt[c,d], Hao
Chen[a]
[a]State Key Laboratory of Biocontrol, School of Ecology, Sun Yat-sen University,
Shenzhen, 518107, China
[b]Fujian Key Laboratory of Pollution Control and Resource Reuse, College of
Environmental Science and Engineering, Fujian Normal University, Fuzhou, 350007,
China
[c]Biosciences Division, Oak Ridge National Laboratory, Oak Ridge, TN, 37831, USA
[d]Department of Microbiology, University of Tennessee, Knoxville, TN, 37996, USA
[*]Corresponding author:
Huaihai Chen, State Key Laboratory of Biocontrol, School of Ecology, Sun Yat-sen
University, Guangzhou, 510006, China; Email: chenhh68@mail.sysu.edu.cn
Jiajiang Lin, Fujian Key Laboratory of Pollution Control and Resource Reuse, College of
Environmental Science and Engineering, Fujian Normal University, Fuzhou, 350007,
China; Email: jjlin@fjnu.edu.cn

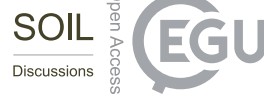

**Abstract**
Understanding the relationship between soil microbial taxonomic compositions and
functional profiles is essential for predicting ecosystem functions under various
environmental disturbances. However, even though microbial communities are sensitive
to disturbance, ecosystem functions remain relatively stable, as soil microbes are likely to
be functionally redundant. Microbial functional redundancy may be more associated with
"broad" functions carried out by a wide range of microbes, than with "narrow" functions
specialized by specific microorganisms. Thus, a comprehensive study to evaluate how
microbial taxonomic compositions correlate with "broad" and "narrow" functional
profiles is necessary. Here, we evaluated soil metagenomes worldwide to assess whether
functional and taxonomic diversities differ significantly between the five "broad" and the
five "narrow" functions that we chose. Our results revealed that compared with the five
"broad" functions, soil microbes capable of performing the five "narrow" functions were
more taxonomically diverse, and thus their functional diversity was more dependent on
taxonomic diversity, implying lower levels of functional redundancy in "narrow"
functions. Co-occurrence networks indicated that microorganisms conducting "broad"
functions were positively related, but microbes specializing "narrow" functions were
interacting mostly negatively. Our study provides strong evidence to support our
hypothesis that functional redundancy is significantly different between "broad" and
"narrow" functions in soil microbes, as the association of functional diversity with
taxonomy were greater in the five "narrow" rather than the five "broad" functions.



**Keywords** Functional redundancy, Soil metagenomics, Functional traits, Taxonomic

compositions,

**1. Introduction**

Microbial communities often exhibit incredible taxonomic diversity, with one gram of

soil harboring millions of microbial species (Gans et al., 2005). However, how such

diversity governs microbial functional potential and ecosystem processes is largely

unknown. Though microbial taxonomic composition is generally sensitive to disturbance

and often do not rapidly recover (Allison and Martiny, 2008), it is unclear how changes

in microbial community composition would regulate ecosystem functioning. Mechanistic

understanding of microbial systems, including microbial taxonomic compositions and

functional potential, is essential for predicting ecosystem functioning under various

environmental disturbances (Torsvik and Øvreås, 2002;Wellington et al., 2003;McGill et

al., 2006).

Though microbial community composition usually shift in response to disturbance,

ecosystem functions could remain relatively stable due to functional redundancy (Allison

and Martiny, 2008). Microbial functional redundancy is an inevitable emergent property

of microbial systems (Louca et al., 2018), as some metabolic functions can be performed

by multiple species, which may thus be substitutable in certain ecosystem processes

(Rosenfeld, 2002), implying that microbial taxonomy and function can be decoupled

(Louca et al., 2016;Louca et al., 2017). Microbial functional redundancy has been mainly

observed in "broad" ecosystem processes (Yin et al., 2000;Rousk et al., 2009;Banerjee et

al., 2016), but is perhaps less significant in "narrow" functions specialized by certain



microorganisms (Schimel, 1995;Balser et al., 2002). However, some studies simulating
microbial diversity reduction and physiological processes challenged the hypothesis of
microbial redundancy in soil microbes (Peter et al., 2011;Philippot et al., 2013;Delgado-
Baquerizo et al., 2016). Such apparent contradictory results suggest the degree of
functional redundancy may depend on the function of interest. Microbes conducting
"broad" metabolic functions, such as carbon decomposition, are likely to distribute across
most taxa (Crowther et al., 2019) and associate with high level of functional redundancy
(Beier et al., 2017;Rivett and Bell, 2018). "Narrow" functions, such as nitrification or
methanogenesis, may be restricted to a few phylogenetic clades (Schimel and Gulledge,
1998), and are hypothesized to exhibit less redundancy than "broad" functions (Schimel,
1995;Rocca et al., 2015). Today, multifunctionality (Hector and Bagchi, 2007) has to be
accounted for to avoid overestimating functional redundancy (Gamfeldt et al., 2008). By
assessing multiple functions, the relationship between microbial diversity and ecosystem
function can be better quantified in the soil (Bastida et al., 2016;Delgado-Baquerizo et
al., 2016).

Nowadays, metagenomics have been increasingly used as a promising comparative

tool (Tringe et al., 2005) to study the relationship between functional and taxonomic
diversities (Fierer et al., 2012a;Fierer et al., 2012b;Fierer et al., 2013;Pan et al., 2014;Leff
et al., 2015;Souza et al., 2015). The growing wealth of soil metagenome data thus poised
well to aid in the generalization of global patterns of microbial attributes and
standardizing frameworks for consistent representation of microbial community (Chen et
al., 2021a;Chen et al., 2021b). However, a synthetic metagenomic analysis to assess how





general microbial taxonomic and functional diversities differ between "broad" and

"narrow" functions across the globe is still lacking.

Here, we constructed soil metagenomic datasets of taxonomic and functional

diversities of five "broad" and five "narrow" functions across seventeen climate zones.

We typically chose SEED Subsystems database (Overbeek et al., 2013) that has diverse

classification at level 1, allowing us to conduct comparison between "broad" versus

"narrow" functions. We selected five "narrow" functions, namely N (Nitrogen

Metabolism), P (Phosphorous Metabolism), K (Potassium Metabolism), S (Sulfur

Metabolism), and Fe (Iron Acquisition and Metabolism). These are typical functional

categories of specific nutrient cycling in Subsystems Level 1 and are only performed by

certain groups of soil microbes (Schimel, 1995). The five "broad" functions selected were

AAD (Amino Acids and Derivatives), CHO (Carbohydrates), CBS (Clustering-based

Subsystems), CVPGP (Cofactors, Vitamins, Prosthetic Groups, Pigments), and Protein

(Protein Metabolism), which are the most abundant functional categories in Subsystems

level 1, and  represent broad-scale functions acquired by a relatively larger group of

diverse soil microbes (Balser et al., 2002). We further contructed the pairswise similarity

of function and taxonomy based on the relative abundance of functional and taxonomic

compositions, respectively, for the five "broad" and the five "narrow" functions. We

hypothesized that the taxonomic similarity of soil microbes would be more linearly

correlated to the functional similarity for the five "narrow" functions in comparison to the

five "broad" functions. Therefore, using these global soil metagenomes, our objective

was to test whether the taxonomic compositions of soil microbes that conduct the five

"narrow" functions are more dependent on the functional compositions, leading to a



lower level of functional redundancy in the "narrow" functions than the "broad"
functions.

**2. Materials and Methods**

**2.1. Data collection**

To ensure that the quality and completeness of the metagenomes analyzed were of
standard, we carefully selected soil metagenomes in MG-RAST server that have been
published in peer-reviewed journals. We searched peer-reviewed publications from 2012
to 2018 from the Web of Science database using search terms such as "soil
metagenome", "shotgun sequencing", and "MG-RAST" to source the metagenomic data
used in this study to their publications. We included soil metagenomes publicly available
in the MG-RAST database that are generated using shotgun sequencing without
amplification or that were directly deposited by peer-reviewed studies into the MG-
RAST database. We then extracted data matrix of taxonomic and functional compositions
of soil metagenomes from MG-RAST public server (https://www.mg-rast.org/) based on
the Study ID and/or MG-RAST ID reported in the publications. Details of each soil
metagenome extracted from publications and MG-RAST database was given in
Supplementary Table S1.
The functional database that we used in this study, SEED Subsystems, is a
categorization system which organizes gene functional categories into a hierarchy with
three levels of resolution (Level 3, 2 and 1) (Overbeek et al., 2013). To download the
taxonomic compositions to soil microbes to conduct "broad" and "narrow" functions, for
each soil metagenome, in the 'Analysis' function of the MG-RAST server



(https://www.mg-rast.org/mgmain.html?mgpage=analysis), we loaded both SEED
Subsystems (Level 3, 2 and 1) as functional profiles and RefSeq (Tatusova et al., 2013)
databases (genus, family, order, class, and phylum levels) as taxonomic compositions
(Chen et al., 2021b). The detailed protocols of MG-RAST server were followed to
analyze the metagenomic functions (Meyer et al., 2008;Wilke et al., 2017). To obtain the
taxonomic compositions of soil microbes that conduct the selected "broad" and "narrow"
functions, we chose 'RefSeq' as source and 'genus' as level, and in 'function filter' we
added the functional categories in Subsystems Level 1 that we are interested in, including
five "broad" functions of AAD (Amino Acids and Derivatives), CHO (Carbohydrates),
CBS (Clustering-based Subsystems), CVPGP (Cofactors, Vitamins, Prosthetic Groups,
Pigments), and Protein (Protein Metabolism), of which the relative abundance was 5-
13%. The functions of AAD, CHO, CBS, CVPGP, and Protein were the most abundant
functional categories in Subsystems Level 1, which were used to represent broad-scale
functions acquired by a large group of diverse soil microbes. Correspondingly, five
"narrow" functions were chosen, namely N (Nitrogen Metabolism), P (Phosphorous
Metabolism), K (Potassium Metabolism), S (Sulfur Metabolism), and Fe (Iron
Acquisition and Metabolism), of which the relative abundance was 0.8-1.4%, as these are
typical functional categories of specific nutrient cycling in Subsystems Level 1 and are
only performed by certain groups of soil microbes. Total hits of taxonomic compositions
of soil microbes conductingeach function at Subsystems Level 1 were calculated as the
sums of hits in different taxonomic categories at RefSeq genus level.

The comparative metagenomic analyses were performed using default settings

(maximum e-value cutoff $= 1e^{-5}$, minimum identity cutoff $= 60\%$, and minimum



alignment length = 50) (Meyer et al., 2008). We then merged the taxonomic compositions
of data matrix of each functions extracted from different studies together to generate new
datasets of microbial taxonomic compositions annotated by the RefSeq database. The
reason why we chose the Subsystems database for functional grouping rather than KEGG
Orthology (KO) (Kanehisa et al., 2015), Clusters of Orthologous Groups (COG)
(Galperin et al., 2014), and Non-supervised Orthologous Groups (NOG) (Huerta-Cepas et
al., 2015) databases was that Subsystems had more diverse classification at Level 1,
allowing us to conduct direct comparison between "broad" versus "narrow" functions.
We chose RefSeq database rather than the traditional ribosomal RNA databases, such as
RDP (Ribosomal Database Project) (Cole et al., 2008), Greengenes (DeSantis et al.,
2006), or Silva LSU/SSU (Pruesse et al., 2007) databases, because taxonomic hits in the
RefSeq database were over 1000-fold higher than the rRNA databases, rendering the
resolution comparable to functional hits for comparison between "broad" and "narrow"
functions. To increase the coverage of our datasets, soil metagenomes with/without
assembly were both included.

The geographic coordinates of latitudes (LAT) and longitudes (LONG) of each soil

metagenome were directly obtained from publications. Based on LAT and LONG,
climate data of mean annual temperature (MAT) and precipitation (MAP) of study sites
for each soil metagenome were extracted from the WorldClim dataset (Fick and Hijmans,
2017) using the R package 'raster' (Hijmans et al., 2015). To examine how microbial
taxonomic diversities of "broad" and "narrow" functions differ globally, soil
metagenomic data was classified into seventeen climate zones based on the main



classification of Koeppen-Geiger Climatic Zones (Kottek et al., 2006) using the R
package 'kgc' (Bryant et al., 2017).

**2.2. Statistical Analyses**
To minimize bias caused by different sequencing depths and read lengths among studies,
we standardize the hits of each taxonomic (or functional) category in each data to relative
abundance by dividing them by the total number of hits. To calculate the pairwise
similarity of taxonomy based on the relative taxonomic abundance at genus level of
microbes conducting the five "broad" and five "narrow" functions, we calculated Bray-
Curtis similarity following log transformation of the compositional taxonomic data by
constructing pairwise Bray-Curtis similarity matrix between each pair of two samples for
each functional categories at Subsystems database at Level 1, which were further
transformed to lists of pairwise Bray-Curtis similarities ordered by sample pair names in
PRIMER 7 (Plymouth Routines in Multivariate Ecological Research Statistical Software,
v7.0.13, PRIMER-E Ltd, UK) (Clarke and Gorley, 2015). To calculate the pairwise
similarity of function, based on the functional abundance at function gene level within
each of the five "broad" and five "narrow" functions, we calculated Bray-Curtis
similarity following log transformation of the compositional functional data by
constructing pairwise Bray-Curtis similarity matrix between each pair of two samples for
each functional categories at Subsystems database at Level 1, which were further
transformed to lists of pairwise Bray-Curtis similarities ordered by sample pair names in
PRIMER 7. To examine the relationship between functional and taxonomic diversities,
Pearson's correlations were constructed between the transformed lists of pairwise Bray-



Curtis similarity of soil metagenomes annotated using Subsystems database at Level 3
(Function) and the RefSeq database at genus level (Taxonomy). The approaches for
processing the relative abundance of compositional data follow the requirements (Gloor
et al., 2017). To analyze the taxonomic composition structures of soil metagenomes
annotated using the RefSeq database at genus level (Taxonomy) of the five "broad" and
five "narrow" functions, PCoA (principal coordinates analysis) and PERMANOVA
(Permutational multivariate analysis of variance) were conducted using the pairwise
Bray-Curtis similarity matrix in PRIMER 7.
To compare microbial taxonomic compositions among the five "broad" and the five
"narrow" functions, one-factor PEMANOVA was conducted using the main test and pair-
wise test in PRIMER 7 with $P$ values and Sq. root reported. Pearson's correlations were
constructed to assess the relationships between functional and taxonomic diversities in
the "broad" and "narrow" functions with adjusted P-Square reported. A RELATE
analysis was also performed to evaluate the relatedness among "broad" and "narrow"
functions by calculating a Spearman's Rho correlation coefficient in PRIMER 7. To
examine the relative abundance of dominant microbial at phylum and class level (mean >
1%) among the five "broad" and five "narrow" functions, heatmaps were constructed
using HeatMapper (Babicki et al., 2016). One-way analysis of variance (ANOVA) with $P$
values adjusted by Bonferroni-correction for multiple comparisons was conducted using
SPSS 22.0 software (Chicago, IL, USA) to evaluate the differences in the relative
abundance of dominant taxonomic compositions (mean > 1%) among climate zones after
the normality of residues and homogeneity of variance were checked using Shapiro-Wilk
and Levene test, respectively. The significance level was set at α=0.05 unless otherwise



stated. To calculate the statistical difference between the relative abundance of dominant
microbial taxonomic groups (mean > 1%) in the "broad" and "narrow" functions, LEfSe
(linear discriminant analysis effect size) method was used
(http://huttenhower.sph.harvard.edu/lefse/) (Segata et al., 2011). Venn's diagrams were
constructed to visualize the amount of dominant microbial taxonomic groups at genus
levels or network nodes shared between the five "broad" and the five "narrow" functions
using InteractiVenn (Heberle et al., 2015).

To find out potential interactions of microbial taxonomic compositions between

"broad" and "narrow" functions across the globe, co-occurrence network analysis was
performed using the Molecular Ecological Network Analyses Pipeline
(http://ieg4.rccc.ou.edu/MENA/) (Zhou et al., 2011;Deng et al., 2012). To make the
minimum observed value close to but no less than 1 as required by the pipeline, the data
of relative abundance were multiplied by $10^6$, which would not change the correlation
coefficients. The data matrix of transformed data matrix was uploaded to construct the
network with default settings, including (1) keeping only the species present in more than
a half of all samples; (2) only filling with 0.01 in blanks with paired valid values; (3)
taking logarithm with recommended similarity matrix of Pearson's correlation
coefficient; and (4) calculation ordered to decrease the cutoff from top using regress
poisson distribution only. A default cutoff value (similarity threshold, $S_t$) for the
similarity matrix was used to assign a link between the pair of species. After that, the
global network properties, the individual nodes' centrality, and the module separation and
modularity were analyzed based on default settings using greedy modularity
optimization. Network files were exported and visualized using Cytoscape software





(Shannon et al., 2003). The scatter plots of within-module connectivity (zi) and among-
module connectivity (Pi) were constructed to show the network node distribution of
module-based topological roles of taxonomic compositions for the "broad" and "narrow"
functions. The threshold values of Zi and Pi for categorizing were 2.5 and 0.62
respectively.

**3. Results and Discussion**
**3.1. Microbial taxonomic compositions**
This study included 845 soil metagenomes across seventeen climate zones around the
world extracted from 56 MG-RAST studies published in 51 peer-reviewed papers. They
resulted in 356090 pairwise comparisons of Bray-curtis similarity in functional
(Subsystems L3) and taxonomic (RefSeq genus) diversities for the five "broad" and five
"narrow" functions, which were analyzed to find out whether the correlations of function
and taxonomy were greater in the five "narrow" functions. Overall, for the five "narrow"
functions, the positive correlations of the pairwise similarity of taxonomy and function
between either two samples ($r^2 = 0.36$-$0.49$) were greater than those for the five "broad"
functions ($r^2 = 0.23$-$0.29$) (Fig. 1). This suggests that rare phylotypes could be more
associated with narrow ecosystem processes than broad-scale functions, supporting the
notion that the abundance of particular specialists could influence narrow functional
measures (Peter et al., 2011;Rivett and Bell, 2018), leading to a lower degree of
functional redundancy associated with "narrow" functions, such as the nutrient cycling
examined in this study.

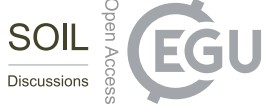

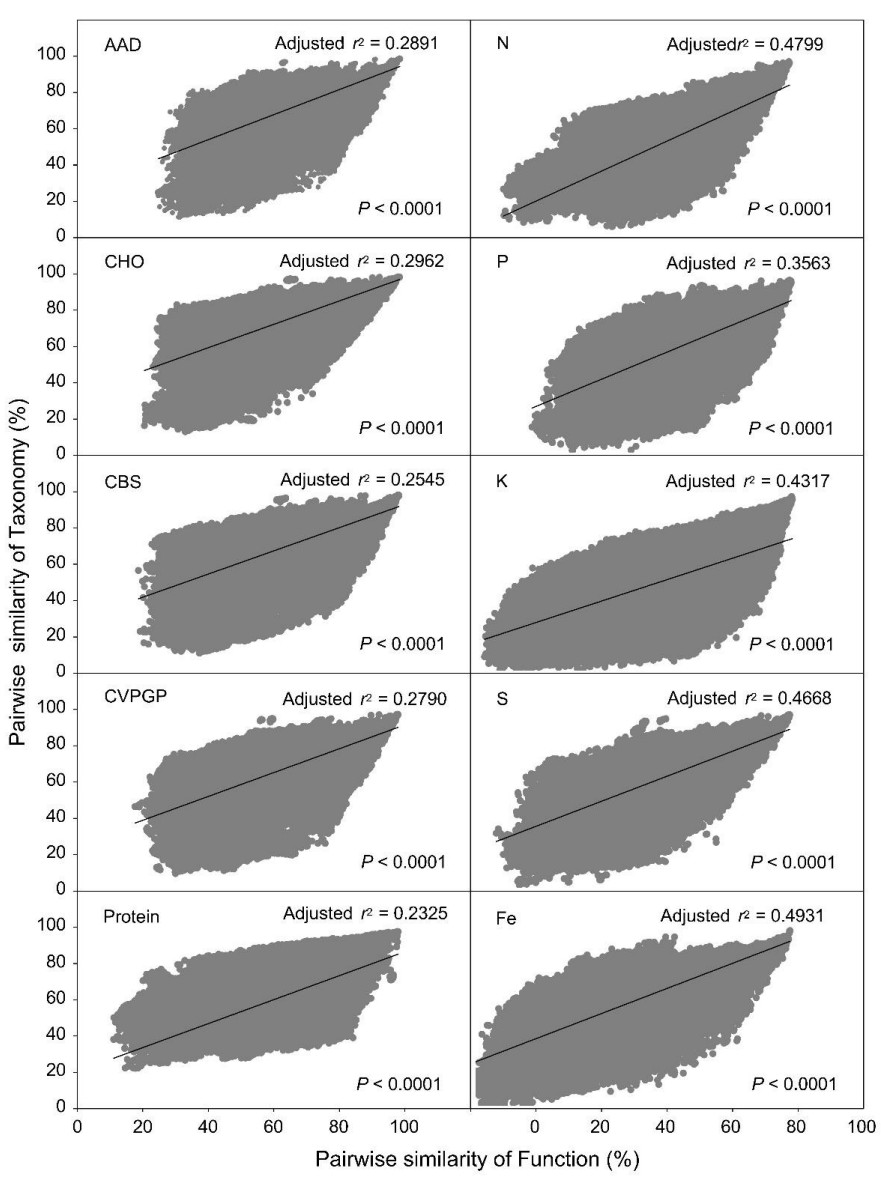

**Fig. 1. Relations between functional and taxonomic beta-diversities for "broad" and "narrow" functions.** Pearson's correlations between pairwise Bray-curtis similarity of microbial taxonomic and functional compositions for "broad" and "narrow" functions annotated using Subsystems at function level (Function) and RefSeq at genus level (Taxonomy). Correlation adjursted $r$-squared and $P$ values are given. "Broad" functions include AAD (Amino Acids and Derivatives), CHO (Carbohydrates), CBS (Clustering-



based Subsystems), CVPGP (Cofactors, Vitamins, Prosthetic Groups, Pigments), and
Protein (Protein Metabolism). "Narrow" functions include N (Nitrogen Metabolism), P
(Phosphorous Metabolism), K (Potassium Metabolism), S (Sulfur Metabolism), and Fe
(Iron Acquisition and Metabolism).

Several soil metagenomic studies have reported a linear relationship between

functional and taxonomic diversities (Fierer et al., 2012b;Fierer et al., 2013;Leff et al.,
2015), indicating a somewhat dependency of microbial functional profiles on taxonomic
compositions. This dependency, however, does not necessarily imply an absence of
microbial functional redundancy. In fact, those studies all showed lower variation of beta-
diversity of metagenomic functions than taxonomy (Fierer et al., 2012b;Fierer et al.,
2013;Pan et al., 2014;Souza et al., 2015) or higher similarity in composition of functional
profiles than taxonomic composition (Leff et al., 2015). Those findings support that
microbial functions are relatively more stable than taxonomy responding to ecological
and environmental perturbations. In this study, the five "broad" and the five "narrow"
functions had relative abundance of 5-13% and 0.8-1.4%, respectively. Thus, the five
"broad" functions are more abundant than the five "narrow" functions. In addition, the
numbers of genes within the categories of the five "broad" functions were also greater
than those of the "narrow" functions. As the diversities of the microbes conducting the
five "broad" functions were also greater than those conducting the "narrow" functions,
we calculated the relationship between the diversities of taxonomy and of function, and
compared these relationships between the five "broad" and the five "narrow" functions.
Our study further evidenced a lower extent of functional redundancy in the five "narrow"
functions compared to the five "broad" functions despite the linear correlations found in
our study.



The boxplots were constructed based on the pairwise similarity of function and
taxonomy to compare similarity ranges of these two compositions related to the five
"broad" functions versus the five "narrow" functions. For the functional compositions at
specific function gene levels, the average similarity of the five "broad" functional
diversity (58%) was comparable to that of the five "narrow" functions (56%) (Fig. 2a).
However, the pairwise similarity of the five "narrow" functions had larger variation, in
which Fe function had the lowest similarity of 36% and N function had the highest
similarity of 69%. On the contrary, the taxonomic similarity of the five "broad" functions
were consistently greater (63-69%) than those of the five "narrow" functions (50-59%).
The PERMANOVA pairwise test was conducted to find out the difference between
taxonomic similarity of microbes conducting the five "broad" and the five "narrow"
functions based on the relative abundance. Our results indicated that the microbial
taxonomic compositions of the five "broad" functions were more phylogenetically
different from those of the five "narrow" functions (13-22%) than from each other (8-
13%) (Supplementary Table 2). The RELATE test was also conducted to evaluate the
relationship of the taxonomic compositions of microbes conducting the five "broad" and
the five "narrow" functions. Our results confirmed that the microbial taxonomic
compositions of the five "broad" functions were more correlated with each other (0.97-
0.99) than those of the five "narrow" functions (0.77-0.94) (Supplementary Table 3).
When the microbial taxonomic compositions of the ten functional categories were
combined in PCoA analysis, the resulting scatter plot showed that the five "broad"
functions were grouped closely together and separated from the five "narrow" functions
(Fig. 2b). Grouping of the ten functions generally explain up to 18.0% of the community



difference, in which the five "narrow" functions were more distinct from each other.
These evidences together suggest that the taxonomic composition of soil microbes
conducting the five "broad" functions were more conserved in taxonomy than those
conducting the five "narrow" functions. However, it should be noted that the current
analysis had some limitations as the metagenomics datasets consisted of sequencing data
that are phylogenetically classified and assigned based on certain the taxonomic and
functional databases. Thus, our results may to some extent depend on the databases
chosen, of which the classification and assignment may not contain potential bias. Future
studies should continue to test this hypothesis using regional samples and individual
datasets.

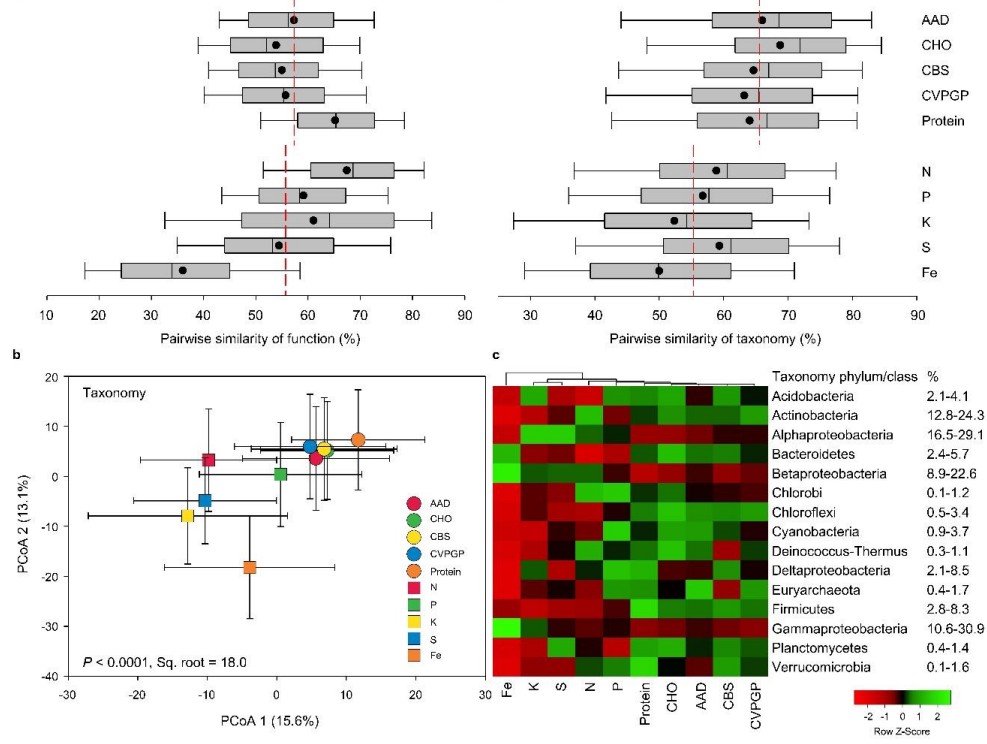




**Fig. 2. Functional and taxonomic diversities for "broad" versus "narrow" functions.**
**a**, Box plots and mean values of pairwise Bray-curtis similarity of microbial functional
and taxonomic diversities for "broad" versus "narrow" functions. **b**, PCoA (Principal
coordinates analysis) showing beta-diveristy of microbial taxonomic diversity for
"broad" and "narrow" functions annotated using RefSeq at genus level (Taxonomy). The
error bars represent the standard deviation of data ranges. Variations (by percentage)
explained by the two principal coordinate dimensions aare given in parentheses. *P* values
and sq. root of PERMANOVA are also given. **c**, Heatmaps showing relative abundance
of dominant microbial taxonomic composition (mean > 0.5%) for "broad" and "narrow"
functions annotated using RefSeq at phylum/class levels (Taxonomy). "Broad" functions
include AAD (Amino Acids and Derivatives), CHO (Carbohydrates), CBS (Clustering-
based Subsystems), CVPGP (Cofactors, Vitamins, Prosthetic Groups, Pigments), and
Protein (Protein Metabolism); "Narrow" functions include N (Nitrogen Metabolism), P
(Phosphorous Metabolism), K (Potassium Metabolism), S (Sulfur Metabolism), and Fe
(Iron Acquisition and Metabolism).

To investigate how microbial taxonomic diversities differ globally, the taxonomic
compositions of soil microbes conducting the five "broad" and the five "narrow"
functions were analyzed among the seventeen climate zones based on the PCoA analysis.
Across climate zones, microbial taxonomic compositions of the five "narrow" functions
(sq. root = 15.2-18.8) were more distinct than the five "broad" functions (sq. root = 13.4-
15.1) based on the PERMANOVA anaysis (Supplementary Fig. 1). This suggests that
microorganisms relating to "broad" functions were similar to each other in taxonomy,
because "broad" functions are more broadly distributed across most taxa, but soil
microbes performing "narrow" functions were more phylogenetically diverse due to the
specialty of "narrow" functions. Thus, though microbial metabolic functions can be
strongly coupled to elemental cycles and certain environmental factors, the decoupling of



microbial taxonomic and functional profiles is still inevitable when a low-dimensional
functional space is projected to a high-dimensional taxonomic space (Louca et al., 2018),
especially for "broad" functions.
The taxonomic compositions of microbes conducting the five "broad" functions were
more abundant in most major phyla, such as Acidobacteria, Actinobacteria,
Bacteroidetes, and Firmicutes, while the relative abundance of the taxonomic
composition of microbes conducting the five "narrow" functions were greater in
Proteobacteria, especially Alphaproteobacteria and Betaproteobacteria (Fig. 2c). Other
studies also found that some bacteria conducting N cycling, such as ammonia-oxidizers
and rhizobia for N fixation, mainly belong to Alphaproteobacteria or Betaproteobacteria
(Stephen et al., 1996;Moulin et al., 2001).

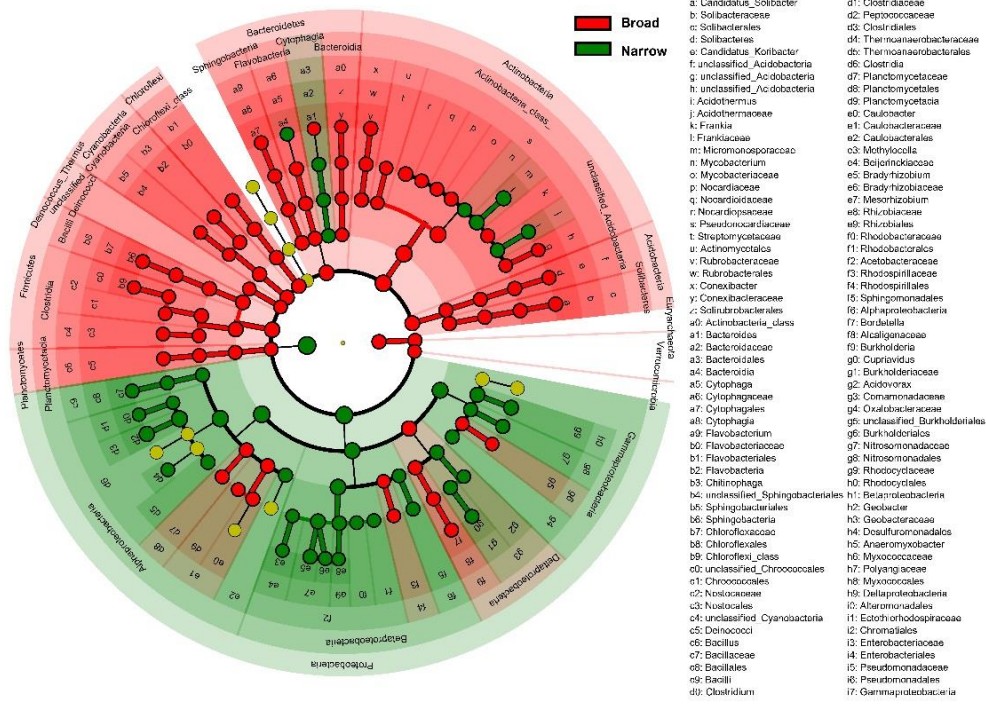




**Fig. 3. Taxonomic compositions shared among "broad" and "narrow" functions**.
Venn's diagrams showing dominant microbial taxonomic groups (mean > 0.1%)
annotated using RefSeq at genus levels (Taxonomy) shared among "broad" and "narrow"
functions.

LEfSe analysis was used to show the dominant microbial groups at the taxonomic

levels of domain, phylum, class, order, family, and genus that were statistically different
between the five "broad" and the five "narrow" functions based on the their relative
abundances. In particular, among the Proteobacteria conducting the five "narrow"
functions, *Bacillaceae* from Bacilli, *Clostridium*, *Peptococcaceae*, and
*Thermoanaerobacteraceae* from Clostridia, *Methylocella*, *Bradyrhizobium*,
*Bradyrhizobiaceae*, and *Rhizobiaceae* from Rhodospirillaceae, and *Cupriavidus* from
Comamonadaceae had higher relative abundance than the others (Fig. 3). The Venn's
diagrams indicated that the taxonomic compositions of soil microbes performing the
"broad" functions shared 68% dominant genera among the five functional categories,
while the proportion was reduced to only 41% for the five "narrow" functions (Fig. 4).
However, it should be stated that all the analyses performed in our study were based on
relative abundance data that is compositional, so it is difficult to directly compare
taxonomic diversities among samples and/or datasets. Despite the differences in the
identification protocol and quantification of soil metagenomes, we deem the effects of
these differences to be trivial for our analyses as we intended to understand the general
patterns of microbial taxonomic and functional linkages, rather than simply compare soil
community structures across samples. By uncovering universal patterns of these
relationships within the microbial community, we can then further establish a potential

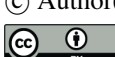


linkage framework to account for the microbial contributions to major biogeochemical
cycles.

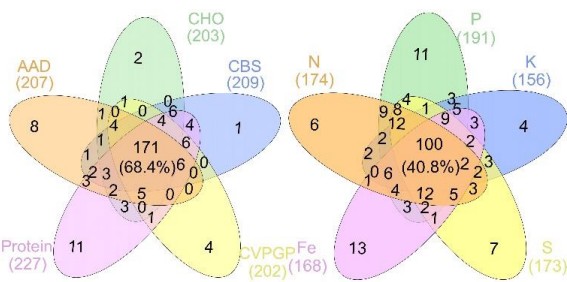


**Fig. 4. Difference of taxonomic compositions between "broad" and "narrow**

**functions".** LEfSe (linear discriminant analysis effect size) results showing the

significant differences in the relative abundance of dominant microbial taxonomic groups

(mean > 0.5%) between "broad" (red) versus "narrow" (green) functions annotated using

RefSeq (Taxonomy). From the center outward, each circle represents the level of domain,

phylum, class, order, family, and genus, respectively. The taxonomic groups with

significant differences are labeled by colors.


Because of functional redundancy of soil microbes, understanding what types of
functions that have more significant association with microbial taxonomy can be critical
for accurate prediction of microbial metabolic activity and flexibility across space and
time. As microbial taxonomic composition and diversity plays critical role in maintaining
ecosystem function (Allison and Martiny, 2008), our results suggest that taxonomic
information alone provides limited utility in predicting basic metabolic capabilities, but
may be capable of forecasting biogeochemical transformations or changes in the rate of
biogeochemical process at ecosystem level (Hall et al., 2018). Investigating functional
redundancy with respect to functions associated with elemental cycles provides useful
information for guiding the development of explicit microbial biogeochemical prediction,



and further delving into major pathways of C and N cycles will be a fruitful approach for
scrutinizing microbes' functional potentials.
**Table 1. Summary of key properties of co-occurrence networks for the five "broad"**
**and the five "narrow" functions.**

| Network Indexes | Total nodes | Total links (positive%) | Average connectivity | Average clustering coefficient | Average geodesic distance | Modularity (modules numbers) |
|---|---|---|---|---|---|---|
| AAD | 225 | 1472 (100%) | 13.084 | 0.663 | 2.873 | 0.695 (11) |
| CHO | 207 | 1155 (99%) | 11.159 | 0.615 | 3.805 | 0.672 (10) |
| CBS | 246 | 1622 (99%) | 13.187 | 0.663 | 2.859 | 0.671 (11) |
| CVPGP | 201 | 1293 (99%) | 12.866 | 0.65 | 3.303 | 0.697 (9) |
| Protein | 285 | 1651 (99%) | 11.586 | 0.638 | 2.992 | 0.749 (14) |
| N | 101 | 519 (12%) | 10.277 | 0.349 | 1.903 | 0.184 (5) |
| P | 160 | 449 (4%) | 5.612 | 0.299 | 3.298 | 0.615 (10) |
| K | 143 | 364 (67%) | 5.091 | 0.08 | 2.676 | 0.429 (6) |
| S | 132 | 264 (15%) | 4 | 0.09 | 2.563 | 0.486 (12) |
| Fe | 95 | 215 (11%) | 4.526 | 0.071 | 2.601 | 0.435 (6) |


**3.2. Microbial taxonomic co-occurrence networks**
Co-occurrence networks of taxonomic compositions were generated to identify potential
interaction patterns of microbial groups that conduct the five "broad" and the five
"narrow" functions across the globe. Network graphs with submodule structures
indicated distinct topology of taxonomic networks between the "broad" and "narrow"
functions (Table 1, Supplementary Fig. 2 and Supplementary Fig. 3). Compared to the
"narrow" functions, the "broad" functions harbored larger and more complex networks
with more nodes (201-285 vs. 95-160) and links (1293-1651 vs. 215-519), with higher
average connectivity (11.2-13.2 vs. 4.0-10.3) and average clustering coefficient (0.64-
0.66 vs. 0.07-0.35). The "broad" function network had 99-100% positive links, while the
"narrow" function had 33-96% negative links.



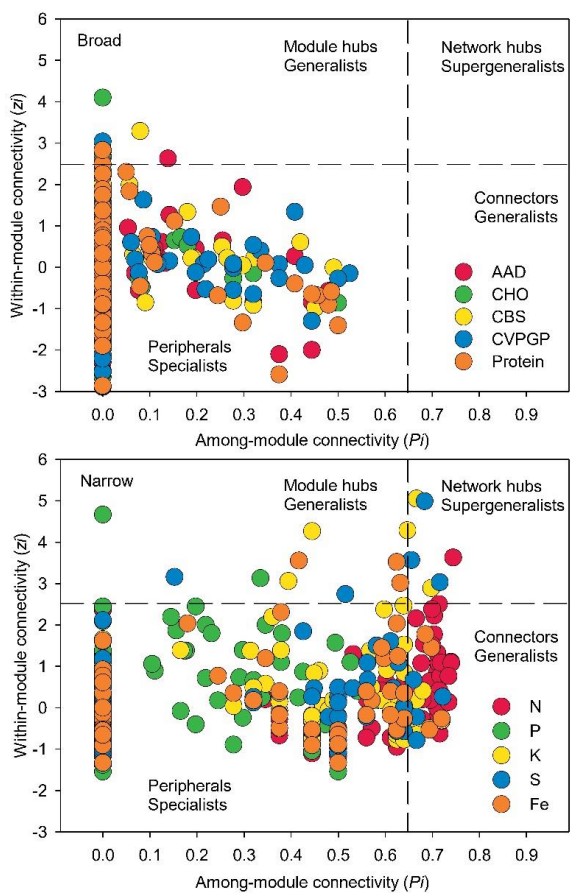

**Fig. 5. Network information of taxonomic compositions for "broad" and "narrow" functions.** Node distribution of module-based topological roles of taxonomic compositions for "broad" and "narrow" functions determined by the scatter plot of within-module connectivity ($z_i$) and among-module connectivity ($P_i$). The threshold values of Zi and Pi for categorizing were 2.5 and 0.62 respectively.

In addition, network modularity was greater in the "broad" functions, indicating that significant correlations between taxonomic compositions of microbes that conduct the five "broad" functions are mainly within similar taxonomic groups. No node could be classfied as connectors in the five "broad" function networks (Fig. 5), reaffirming that the



"broad" function networks had links mainly within modules of similar species. In the co-
occurrence network of taxonomic composition of the "narrow" functions,  13% of the
nodes were identified as connectors linking several modules (high $Pi$) connectors, while
3% were identified as module hubs that connected other nodes within their own modules
(high $Zi$), indicated by the $Zi$-$Pi$ plot (Olesen et al., 2007;Deng et al., 2012). Thus,
significantly less nodes were identified as module hubs in the co-occurrence network of
the taxonomic composition of the "broad" functions, indicting less correlations found
among different modules. This is expected given that module was comprised of genera
that were mainly from the same phylogenetic groups. This difference was consistent with
the Venn's diagrams showing significantly more nodes (54%) shared among the five
functional categories representing the "broad" functions, while only 5% of the nodes
were overlaid among the five "narrow" function networks (Fig. 6). Environmental
conditions likely determine the microbial taxonomic composition, and microbial
phylotypes sharing similar habitat preferences tend to co-occur (Delgado-Baquerizo et
al., 2018;Ram ŕez-Flandes et al., 2019). We emphasize that this analysis is a combination
of snapshots of microbial communities compared across space, thus environmental
conditions (at the same geographic location) may vary, and the levels of functional
redundancy may change depending on the mechanisms selecting specific functions and
the phylogenetic distribution of those functions (Louca et al., 2018).

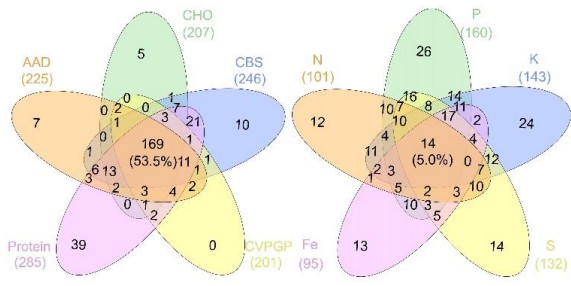





**Fig. 6. Taxonomic network nodes shared among "broad" and "narrow" functions**.
Venn's diagrams showing the microbial taxonomic network nodes shared among "broad"
and "narrow" functions.

**3.3. Conclusion**
By analyzing and generalizing microbial taxonomic and functional profiles, we provide
strong evidence that the degree of soil microbial functional redundancy differ
significantly between "broad" and "narrow" functions across the global. The level of
functional redundancy varies depending on the functions of interest. Here, by contrasting
the five "broad" metabolic functions and the five "narrow" functions that are important
for elemental cycles, we found lower levels of functional redundancy associated with the
five "narrow" functions of biogeochemical cycling, despite the fact that even for the five
"narrow" functions, there is still a high level of functional redundancy in the soil
communities. Although there is a caveat concerning direct comparison of metagenomic
data, the present study demonstrated the use of comparative metagenome and co-
occurrence network analysis in generalizing patterns of microbial characteristics
regulating biogeochemical cycling of major elements. With the increasing advancement
of sequencing techniques and data coverage, future sequencing efforts will likely increase
our confidence in comparative metagenomes and provide time-series information to
further identify to what extent microbial functional redundancy regulates dynamic
ecological fluxes across space and time.

**Author Contributions**



Huaihai Chen conceived the study, performed the data analysis, interpreted the results,
and drafted the manuscript. CL, YY, CWS, and Hao Chen secured the research funding.
KM, YH, YY, and Hao Chen critically assessed and interpreted the findings. All authors
discussed results, commented on, edited, revised, and approved the manuscript.

**Funding**
This study is supported by the National Natural Science Foundation of China
(31872691), Basic and Applied Basic Research Foundation of Guangdong Province.

**Acknowledgements**
We thank the authors of the publications included in our study, without which this global
metagenomic analysis would not be possible.

**Data Availability Statement**
The data that support the findings of this study are available from the corresponding
author upon request. All metagenomic data used in this study are publicly assessable in
the MG-RAST server with study and MG-RAST ID reported in supplementary files.

**Competing Interests**
The authors declare no competing interests.

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

ecological network of soil microbial communities in response to elevated CO2.

*MBio* 2**,** e00122-00111.
