# Peer review of "Lower functional redundancy in "narrow" than "broad" functions in global soil 1 metagenomics 2 3 Huaihai Chena,\*, Kayan Maa, Yu Huanga, Jiajiang Linb, Christopher W. Schadtc,d, Hao 4 Chena 5 6 7 aState Key Laboratory of Biocontrol, S"

_SOIL, 2021_

## Author Response (AR1)

Reply to Editor

Thank you so much for giving us the opportunity to revise our manuscript. After careful revision, we believe that our revised manuscript has completely addressed each reviewer's questions point by point. If any question, please let us know. Thanks so much.

Reply to Referee #1

**Major comments:**
   1. **Better defining the concept of functional redundancy in the introduction:**
Done. Thanks so much for the comment. We have added the two different definition of functional redundancy (strict and partial) with the reference as suggested by the reviewer (Pg3 L66-Pg4 L71).

   2. **Mentioning functional redundancy of non-metabolic processes in the introduction:**
Done. Thanks very much for the suggestion. We have also added more information regarding functional redundancy of non-metabolic processes in the introduction section as suggested by the reviewer (Pg4 L71-76).

   3. **L367 Mention the inevitability of function redundancy sooner in the introduction:**
Done. Thanks so much for the suggestion. We have mentioned the inevitability of microbial function redundancy earlier in the introduction section as suggested by the reviewer (Pg4 L83-84).

   4. **L72-73 Mention the limitations in measuring the factors controlling niche space:**
Done. Thanks so much for the suggestion. We have also added more information regarding the possible reasons for our current contradictory understanding of functional redundancy as suggested by the reviewer (Pg4 L84-88).

   5. **L254-255 Add reference for the threshold values in the network analysis:**
Done. Thanks so much for the comment. We have added the references for choosing the threshold values of network connectivity profiles (Pg13 L276-277).

**Technical Corrections:**
   1. **L54 Change to "does not rapidly recover":**
Done. Thanks for the comment. The word has been changed as suggested by the reviewer (Pg3 L54).

   2. **L60 Change to "shifts":**
Done. Thanks for the comment. The word has been changed as suggested by the reviewer (Pg3 L60).

   3. **L106 Change to "pairwise":**
Done. Thanks for the comment. The word has been changed as suggested by the reviewer (Pg6 L120).

   4. **L333 Should state "based on certain taxonomic and functional databases":**

Done. Thanks for the comment. The word "the" has been deleted as suggested by the reviewer (Pg17 L355).

   5. **L335 Should be "may contain potential bias":**
Done. Thanks so much for the comment. The word "not" has been deleted as suggested by the reviewer (Pg17 L357).

   6. **Captions for Figure 3 and 4 need to be switched.:**
Done. Thanks so much for the comment. The captions for Figure 3 and 4 have been switched.

Reply to Reviewer #2

**Major comments:**
   1. **Creating an overview figure highlighting how the data was input, transformed, and analyzed:**
Done. Thanks so much for the suggestion. We have added a figure of an overview of data acquisition, transformation, and analysis processes in this study in the supplementary files as suggested by the reviewer (Pg13 L276-277, Supplementary Fig. 1).

   2. **L206 Deal with unclassified groups in taxonomic classification levels across datasets:**
Done. Thanks for the comment. We used the genus level as the taxonomic classification level across different datasets. Following default setting in MG-RAST, if the species were classified into the higher classification levels than genus but failed to be identified at the genus level, they were classified into "unclassified" groups. Across different studies, there were $2.16 \pm 0.85$ % of sequences belonging to the "unclassified" groups, showing that most taxonomic groups could be classified into the genus level. We have added this information in the Materials and Methods section as suggested by the reviewer (Pg8 L169-175).

   3. **L437-438 More discussion about the differences in network properties between broad and narrow functions:**
Done. Thanks for the suggestion. We have added more information in the discussion section about the implications of the differences in network properties between broad and narrow functions as suggested by the reviewer (Pg23 L465-472).

   4. **Discussion about the relationship may be more coupled in extreme environments with significant selective pressure:**
Done. Thanks for the comments. We agreed with the reviewer that certain environmental factors may have significant effects on the coupling of taxonomy and function due to their selective pressure, such as the extreme environment of ice cap, so we have added this point in the discussion section that different protocols and analyses from collected studies may lead to potential bias in the discussion and conclusion (Pg19 L390-394).

**Technical Comments:**
   1. **L106 Change to "constructed":**
Done. Thanks for the comment. The word has been changed as suggested by the reviewer (Pg6 L120).

2. **L156 Change to "conducting each":**

Done. Thanks for the comment. The word has been changed as suggested by the reviewer (Pg8 L175).

3. **L305 Adding/changing the topic sentences:**

Done. Thanks for the comment. The sentences have been rephrased as suggested by the reviewer (Pg16 L327-329, Pg20 L412-415, Pg22 L455-458).